# *Bartonella* Associated Cutaneous Lesions (BACL) in People with Neuropsychiatric Symptoms

**DOI:** 10.3390/pathogens9121023

**Published:** 2020-12-04

**Authors:** Edward B. Breitschwerdt, Julie M. Bradley, Ricardo G. Maggi, Erin Lashnits, Paul Reicherter

**Affiliations:** 1Intracellular Pathogens Research Laboratory, Comparative Medicine Institute, College of Veterinary Medicine, North Carolina State University, Raleigh, NC 27607, USA; julie_bradley@ncsu.edu (J.M.B.); rgmaggi@ncsu.edu (R.G.M.); ewlashni@ncsu.edu (E.L.); 2Dermatology Clinic, Truman Medical Center, University of Missouri Kansas City, Kansas City, MO 64108, USA; drpderm@msn.com

**Keywords:** infection, bacteria, psychoses, transmission, stretch marks, bartonella, neuropsychiatric, cutaneous lesions, BACL

## Abstract

*Bartonella* species are globally important emerging pathogens that were not known to infect animals or humans in North America prior to the human immunodeficiency virus (HIV) epidemic. Ongoing improvements in diagnostic testing modalities have allowed for the discovery of *Bartonella* species (spp.) DNA in blood; cerebrospinal fluid; and the skin of patients with cutaneous lesions, fatigue, myalgia, and neurological symptoms. We describe *Bartonella* spp. test results for participants reporting neuropsychiatric symptoms, the majority of whom reported the concurrent development of cutaneous lesions. Study participants completed a medical history, a risk factor questionnaire, and provided cutaneous lesion photographs. *Bartonella* spp. serology and *Bartonella* alpha proteobacteria enrichment blood culture/PCR were assessed. Within a 14-month period, 33 participants enrolled; 29/33 had serological and/or PCR evidence supporting *Bartonella* spp. infection, of whom 24 reported concurrent cutaneous lesions since neuropsychiatric symptom onset. We conclude that cutaneous lesions were common among people reporting neuropsychiatric symptoms and *Bartonella* spp. infection or exposure. Additional studies, using sensitive microbiological and imaging techniques, are needed to determine if, or to what extent, *Bartonella* spp. might contribute to cutaneous lesions and neuropsychiatric symptoms in patients.

## 1. Introduction

Fever, lymphadenopathy, and a history of a cat scratch/bite are the prototypical manifestations of Cat Scratch Disease (CSD), an acute onset illness caused by *Bartonella henselae* [1,2]. A cutaneous primary inoculation papule is also reported in 60–90% of CSD cases [3]. Although rare, a spectrum of other dermatological lesions, including maculopapular and urticarial eruptions, granuloma annulare, erythema nodosum, erythema marginatum, thrombocytopenic purpura, leukocytoclastic vasculitis, multiple granulomatous lesions, and erythema annulare, have been reported in CSD patients [3]. Thus, a spectrum of cutaneous lesions has been historically associated with CSD.

With the advent of increasingly sensitive diagnostic modalities, chronic blood stream infection with *Bartonella* spp. has been reported in patients with cardiovascular, neurological, and rheumatological diseases [1,2,4]. If, or to what extent, persistent *Bartonella* spp. infection might result in cutaneous manifestations in patients with neuropsychiatric symptoms has not been determined. To our knowledge, linear cutaneous lesions were first clinically associated with *B. henselae* infection in 2002 in 10 pediatric patients with a history of abdominal pain, skin rash, mesenteric adenitis, gastritis, and duodenitis following a cat scratch or tick bite [5]. These cutaneous lesions were described as purpuric, serpiginous, nodular rashes that did not blanche when pressure was applied. Subsequently, cutaneous linear lesions, referred to as “striae”, were described as “prototypical for bartonellosis” in a book published in 2008 [6]. In 2013, we described *B. henselae* infection in a boy whose symptoms included linear cutaneous lesions—at the time, also referred to as “striae”—as well as headaches, memory loss, disorientation, and peripheral neuropathic pain; *B. henselae* deoxyribonucleic acid (DNA) was amplified and sequenced from a biopsy of one cutaneous lesion, as well as several blood and serum specimens [7]. In 2019, we described *B. henselae* blood stream infection in another boy diagnosed with Pediatric Acute-onset Neuropsychiatric Syndrome (PANS) who had skin lesions described as “Bartonella-associated cutaneous lesions (BACL)”. In this patient, the cutaneous lesions and neuropsychiatric symptoms resolved with antimicrobial therapy [8]. In 2020, three additional children with neuropsychiatric symptoms and concurrent cutaneous lesions, referred to as “Bartonella tracks”, were described [9]. Due to variability in the cutaneous lesions in patients with acute-onset CSD and in those with chronic symptoms and blood stream infections, we propose the more inclusive designation encompassing historical terms (striae and tracks) to describe the overall spectrum of lesions as Bartonella-Associated Cutaneous Lesions (BACL).

In retrospect, it is unfortunate that the term “striae” was initially applied to lesions reported in association with clinically suspected or microbiologically confirmed cases of Bartonellosis. In a 2018 retrospective case series of 12 adolescent boys with striae distensae, the authors concluded that horizontal striae distensae of the lower back in adolescent boys is associated with a rapid growth spurt, tall stature, and family (genetic) history of striae distensae [10]. As only one boy was tested for CSD, the authors stated that the possibility of infection as a trigger for striae distensae could not be excluded based on their study. Although genetics can influence patterns of disease expression in family units, common environmental and vector exposures can also result in familial disease patterns. As summarized in association with a recent familial cluster, familial infection with *Bartonella* spp. has been reported in families residing in Australia, Canada, Denmark, the Netherlands, and the United States [11].

The purpose of this investigation was to describe the largest case series to date involving microbiological testing for Bartonelloses in individuals with self-reported neuropsychiatric symptoms, the majority of whom developed concurrent cutaneous lesions. Dermatologists, neurologists, and psychiatrists should be aware that *Bartonella* exposure/infection occurs in patients reporting neuropsychiatric symptoms and a spectrum of cutaneous lesions.

## 2. Results

### 2.1. Study Design and Subject Enrollment

Between August 2018 and October 2019, participants experiencing neurological or neuropsychiatric symptoms were enrolled after contacting the investigators requesting inclusion in a research study entitled “Detection of *Bartonella* Species in the blood of healthy and sick people”. A cross-sectional study was performed to determine the seroprevalence to six *Bartonella* species/genotypes. Bacteremia was concurrently assessed by means of a *Bartonella* alpha-Proteobacteria growth medium (BAPGM) enrichment blood culture platform. Twenty-nine of 33 participants with self-reported neuropsychiatric symptoms and serological and/or PCR evidence supporting *Bartonella* spp. infection were included in the study. There were 12 males and 17 females, ranging in age from 12 to 58 years. The mean age for males and females was 23.3 years and 28.1 years, respectively. Fourteen participants were students (two veterinary students). Nine students were adolescents (14 to 17 years old, three of which were medically unable to attend school); the remaining five students were 21 to 29 years old. Eight participants were veterinary workers (veterinarians or technicians). One participant each was a landscaper, teacher, and psychologist. The remaining four participants, all in their 20s, were medically disabled. Twenty-eight participants were from the United States, residing in sixteen states. One participant only lived in Australia. Only 10/24 participants reported consulting a dermatologist, whereas 18/29 and 21/29 reported consultation with a neurologist or psychiatrist, respectively. Participants reported median illness durations of 57.7 months (range four months to 13 years).

### 2.2. Reported Neuropsychiatric Symptoms

The frequencies of self-reported neuropsychiatric symptoms are summarized in Table 1. Sleep disorders, mental confusion, irritability/rage, anxiety, depression, and headache/migraine were predominant symptoms. Other symptoms, not attributable to the skin or nervous system, are also listed in Table 1. 

### 2.3. Reported Cutaneous Lesions

Twenty-four of 29 participants (83%) reported cutaneous lesions during their illness. Ten participants had only diffuse skin eruptions, 2 only linear lesions, and 12 had both diffuse skin eruptions and linear lesions. Linear lesions were characteristically serpiginous pink-to-red patches and/or depressed or slightly elevated plaques (Figure 1, Figure 2, Figure 3, Figure 4, Figure 5, Figure 6 and Figure 7). Seven participants had urticarial papules and plaques with dermatographism (Figure 8). A veterinarian, scratched by a cat, developed recurrent, small, discrete pustules around the scar (Figure 9). Linear lesions occurred in five males, all of whom were 14 or 15 years old, and in four females after puberty, who ranged in age from 14 to 17 years old. When questioned about risk factors for striae distensae, 22 of 24 successfully contacted participants reported no bodybuilding activities; one male reported a rapid weight gain, and five participants reported prior treatment with prednisone, four of whom were treated for one month or less. Marfans disease and Cushing’s syndrome were not listed as diagnoses on the questionnaire; however, one woman reported testing negative for Cushing’s syndrome. Two women reported a diagnosis of Hashimoto’s thyroiditis. Linear lesions were not visualized in the photographs provided from three previously pregnant woman, all of whom reported rashes.

### 2.4. Bartonella spp. Seroprevalence and Blood Stream Infection

Based on the case definition, all 29 participants were *Bartonella* spp. indirect fluorescent assay (IFA) seroreactive and/or polymerase chain reaction (PCR)-positive (Table 2). Sixteen participants were both PCR-positive and IFA seroreactive to one or more *Bartonella* spp. antigens. Seven participants were *Bartonella* spp. seroreactive but PCR-negative, and six participants were PCR-positive but not seroreactive. For the 23 seroreactors, IFA titers were 1:64, 1:128, or 1:256 for 13, 4, and 6 participants, respectively. 

### 2.5. ddPCR Results on Direct Blood Testing versus BAPGM Enrichment Blood Culture

Of the 21/29 ddPCR-positive participants, one was positive only from blood; 13 were positive only after BAPGM enrichment culture (7, 14, or 21 days); and 7 were positive from both blood and BAPGM enrichment culture (Table 3). The participant infected with *B. koehlerae* (Figure 9) was blood DNA extraction qPCR-positive and ddPCR-positive from one and two collection dates, respectively, but BAPGM enrichment blood culture DNA extractions were qPCR and ddPCR-negative for all three blood collection dates and all three incubation testing time points. 

Overall, 22 participants were PCR-positive, four by quantitative polymerase chain reaction (qPCR) and 21 by droplet digital polymerase chain reaction (ddPCR). Samples that were qPCR/DNA sequence-positive exhibited a Ct value (calculated cycle number at which the PCR product crosses a threshold of detection) ranging from 35.8 to 41.2, indicating a very low bacteria gene load in blood or enrichment blood cultures for the four qPCR+ samples (these values represent between one and 0.05 bacteria gene target copy per microliter of sample). Samples positive by ddPCR also showed a very low bacterial gene load. The number of *Bartonella*-positive ddPCR droplets varied between one and three per sample. All qPCR and ddPCR (no droplets visualized)-negative controls remained negative throughout the study. 

Based upon DNA sequencing, three participants were infected with *B. henselae* and one with *Bartonella koehlerae* (Table 2). Three of four qPCR-positive patients were ddPCR-positive from blood or BAPGM enrichment blood culture. qPCR DNA sequences were analyzed by BLAST. Three amplicons shared sequence identities of 109/110 bp (99.1%) with the *B. henselae* Houston I strain (GenBank CP020742) or 110/110 bp (100%) with the *B. henselae* San Antonio 2 (SA2) strain (GenBank AF369529). A longer sequence could be amplified from one of the three *B. henselae* qPCR-positives, which were 549/549-bp (100%) homologous with *B. henselae* SA2 (GenBank AF369529). The *B. koehlerae* amplicon shared 100% sequence identity (110/110 bp) with *B. koehlerae* (GenBank accession AF312490). 

### 2.6. Animal and Vector Exposures

As reported in their medical surveys, the 29 participants included in the study described the following animal exposures: dogs *n* = 27, cats *n* = 22, pet birds *n* = 8, horses *n* = 7, pet reptiles *n* = 9, pet rabbits *n* = 11, pet rodents *n* = 16, cattle *n* = 5, goats *n* = 3, poultry *n* = 6, and swine *n* = 4. Pets slept in beds with 19/29 participants, and 23/29 participants reported allowing pets to lick their faces and hands. None of the animals to whom exposure was reported were tested for *Bartonella* spp. Reported known vector exposures included bed bugs *n* = 3, biting flies *n* = 17, fleas *n* = 23, lice *n* = 15, mosquitos *n* = 29, scabies mites *n* = 10, spiders *n* = 20, and ticks *n* = 26. None of the vectors to whom exposure was reported were tested for *Bartonella* spp.

## 3. Discussion

This case series describes 29 people with self-reported neuropsychiatric symptoms and evidence of *Bartonella* spp. exposure or infection, the majority of whom (83%) had cutaneous lesions accompanying their illness onset. Combining an enrichment blood culture with ddPCR substantially improved the detection of *Bartonella* spp. DNA in this study. Diagnostically, ddPCR is a powerful molecular technique that uses a water–oil emulsion technology driven by microfluidics and surfactant chemistry to massively partition samples into 15,000 to 20,000 1-ηL-sized droplets prior to performing DNA amplification [12]. DNA targets within the original sample are randomly localized within these droplets—after which, the amplification of DNA within each drop is recorded using fluorescently labeled probe detection. The number and fluorescent output for each droplet is read in a manner similar to flow cytometry, where each individual droplet is then identified as being positive or negative for the template (pathogen gene target) of interest. 

The frequency and extent to which *B. henselae* or other *Bartonella* spp. can induce cutaneous lesions or neuropsychiatric symptoms remains unclear. However, there is increasing case-based data to support an association between *Bartonella* spp. and cutaneous lesions, including urticarial papules and plaques along with erythematous, serpiginous patches, and plaques, reported as BACL in this case series. Maculopapular rash [13,14], septal panniculitis [14,15], and vasculitis [16,17] have been reported previously in association with *Bartonella* spp. infections and cutaneous lesions.

In the context of chronicity, histopathology confirmed mild, nonspecific lymphoplasmacytic infiltrates in lesional cutaneous biopsies, consistent with a chronic antigenic immune response in two *B. henselae* case reports [7,8]. As depicted in Figure 1, Figure 2, Figure 3, Figure 4, Figure 5, Figure 6 and Figure 7, the cutaneous lesions we refer to as linear BACL are often vertical, can be narrow or very wide, and do not occur in areas such as the back, where rapid growth would typically contribute to horizontal striae. No participant with linear BACL in this case series reported an association with conditions with known associations with striae distensae, including pregnancy, rapid growth, weightlifting, Cushing’s syndrome, or Marfans disease. Linear BACL lesions did not develop in a 36-year-old veterinarian infected with *B. henselae*, despite treatment for environmental allergies/rash with prednisone for one year prior to study entry. However, in the context of medical complexity, she was examined by 11 medical specialists, including an allergist, dermatologist, and neurologist.

Previously, we proposed that dogs may serve as sentinels for human exposure to *Bartonella* spp. [18,19]. Additionally, dogs develop similar, if not identical, pathology when persistently infected with a *Bartonella* spp. [20]. In the context of a potential common exposure source, infection with *B. henselae* was documented in an elderly man with ulcerated nodular skin lesions and in his dog with histopathologically confirmed panniculitis [21]. Identical SA2 strain types of *B. henselae* DNA were amplified and sequenced from the man’s enrichment blood culture and from the dog’s panniculitis biopsy. Ulcerative skin lesions and *B. henselae* infection were also documented by PCR amplification and DNA sequencing in a beagle with vasculitis and superficial erosions of the ear tip [22]. These and other disease associations shared by dogs and humans suggest that a comparative infectious disease approach might enhance the clinical understanding of these stealth bacteria in a timelier manner [1,23]. 

Based upon questionnaire responses, animal and vector exposures were reported frequently by study participants. The primary mode of *Bartonella* spp. transmission is via arthropod vectors, including, but not limited to, fleas, lice, sandflies, and ticks [1,2,4]. Other modes of transmission include animal bites and scratches, blood transfusions, and needle sticks. As further illustrated by this study, veterinary workers and others with frequent exposure to arthropods and animals may be occupationally at risk for acquiring infection and, thus, a sentinel population to enhance disease understanding [24]. To facilitate the subsequent acquisition by blood-sucking arthropods, vector-borne organisms evolved to avoid immune elimination, to localize in cutaneous tissues, and to downregulate the local inflammatory response, thereby avoiding the induction of obvious skin lesions [24]. Despite evolutionary adaptations by vectors and vector-borne organisms, skin lesions occur in association with a variety of vector-borne diseases. Three somewhat prototypical examples of acute-onset cutaneous lesions induced by tick-transmitted vector-borne pathogens are: erythema chronicum migrans (ECM) associated with *Borrelia burgdorferi* infection [25], the maculopapular rash with Rocky Mountain spotted fever [26] caused by *Rickettsia rickettsii* and ehrlichiosis [26] caused by *Ehrlichia chaffeensis* and *Ehrlichia ewingii*. In contrast, acrodermatitis chronica atrophica (ACA), most commonly caused by *Borrelia afzelii,* occurs months to years after infection [27]. Since *B. henselae* and *B. koehlerae* are predominantly flea-borne pathogens, flea (and other arthropod) exposures should be a component of each patient’s dermatological and neuropsychiatric histories. In addition, when dermatologists consult with families experiencing flea infestations, *Bartonella* spp. testing should be considered if symptoms subsequently develop during the ensuing months or years.

Blood stream infection with *Bartonella* spp. has been recently reported in blood donors and other healthy individuals [28,29]. Thus, despite PCR documentation of *Bartonella* spp. DNA in blood or enrichment blood cultures from the majority (22/29) of the study participants, it is not possible to confirm causation for the neuropsychiatric symptoms or the cutaneous lesions. Controlled treatment trials are likely necessary to determine if or to what extent infections with *Bartonella* spp. contribute to the clinical presentations described in this case series. The resolution of skin lesions and neuropsychiatric symptoms with antimicrobial therapy, as described in the boy diagnosed with PANS [8], provided potential support for causation. Similar to previous studies [28,29,30], a subset (6/29 in this study) of blood stream PCR-positive individuals were not *Bartonella* spp. seroreactive. The reason for this discrepancy remains unclear. The detection of antibody reactivity only indicates exposure to one or more *Bartonella* spp.; however, seronegative infection is not unusual. Thus, serology, enrichment blood culture, and PCR should be used concurrently prior to and during the treatment of suspected Bartonelloses patients.

The limitations of this study included participant self-selection and self-reporting, as these factors may have positively biased the findings. Participant age, duration and severity of illness, and prior treatment regimens likely influenced the microbiological serology and enrichment blood culture/PCR results. Due to the individual’s concerns over neuropsychiatric symptoms, most participants did not consult a dermatologist; nonprofessional photographs were provided by participants or parents, and skin biopsies were not obtained for histopathology, *Bartonella* PCR, or immunohistochemical imaging of *Bartonella* organisms.

Based upon the limited case reports and this case series, it is imperative to design studies that will determine if, or to what extent, *Bartonella* spp. might contribute to concurrent cutaneous lesions in patients with neuropsychiatric symptoms. Mechanistically, could long-standing microvascular injury induced by this endotheliotropic bacteria result in chronic vascular inflammation, local mast cell activation, collagen injury, and the development of a spectrum of BACL [31,32]? As *Bartonella* spp. are predominantly vector-borne pathogens that can localize to the skin, dermatologists should assess vector and animal exposures in patients with unusual or unexplained cutaneous lesions and consider testing patients with BACL and neuropsychiatric symptoms for *Bartonella* spp. infection.

## 4. Materials and Methods

### 4.1. Study Enrollment 

Between August 2018 and October 2019, participants experiencing neurological or neuropsychiatric symptoms were enrolled after contacting investigators requesting inclusion in a research study entitled “Detection of *Bartonella* Species in the blood of healthy and sick people.” A cross-sectional study was performed to determine the seroprevalence to six *Bartonella* species/genotypes. Bacteremia was concurrently assessed by means of a *Bartonella* alpha-Proteobacteria growth medium (BAPGM) (IPRL, NCSU, Raleigh, NC, USA) enrichment blood culture platform. Participants were considered to have *Bartonella* spp. exposure or infection if they were *Bartonella* spp. seroreactive at an IFA titer of ≥1:64, qPCR or ddPCR positive, or positive by more than one testing modality.

### 4.2. Questionnaire Data and Blood Specimen Collection

A standardized questionnaire, including demographic information, symptoms experienced, domestic and wild animal bites, scratches or exposures, and travel history, was completed. Exposure to, or bites by, different arthropods (lice, fleas, ticks, mites, bed bugs, and others) was recorded. This study was carried out in accordance with all relevant guidelines applying to human study participation. Most participants were enrolled after publication of the case report involving the boy diagnosed with PANS [8]. In the case of children, the parents requested enrollment and provided signed permission for testing. All participants (or parents) completed a detailed questionnaire used in previous collaborative studies from our laboratory [30]. 

Only participants with positive *Bartonella* spp. serological or molecular test results were included in the study. Participants/parents provided cutaneous lesion photographs for dermatologist review and written permission for publication. The dermatological description of cutaneous lesions was determined by the dermatologist author. Striae distensae risk factors were determined retrospectively via an email questionnaire provided to all participants reporting cutaneous lesions.

Blood was collected aseptically into ethylenediaminetetraacetic acid (EDTA) anti-coagulated and serum separator tubes for shipment to the Intracellular Pathogens Research Laboratory (IPRL), Comparative Medicine Institute, College of Veterinary Medicine, North Carolina State University (NCSU) for Bartonella testing. With one exception (two collection dates), each participant provided three blood and serum specimens collected on alternate days to increase the *Bartonella* spp. enrichment blood culture/PCR sensitivity [33]. Approximately 10–12 mL of blood (5–6 mL in EDTA anti-coagulant tubes and 5–6 mL in serum separator tubes) was collected every other day for three time points at the time of enrollment. Aseptic blood collection, including chlorhexidine decontamination of the skin, was performed by an experienced nurse. 

### 4.3. Bartonella IFA Serological Testing

As described in previous studies from our research group [11,30,34], each participant was tested using six IFA assays, each representing a unique *Bartonella* species or subspecies. *Bartonella vinsonii* subsp. *berkhoffii, B. henselae, B. koehlerae*, and *B. quintana* antibodies were determined in the IPRL at NCSU (Raleigh, NC, USA) using cell culture-grown bacteria as antigens and following standard IFA techniques [30,34]. Canine isolates of *B. vinsonii* subsp. *berkhoffii* genotype I (NCSU 93CO-01 Tumbleweed, ATCC type strain #51672), *B. vinsonii* subsp. *berkhoffii* genotype II (NCSU 95CO-08, Winnie), and *B. vinsonii* subsp. *berkhoffii* genotype III (NCSU 06CO-01 Klara); feline isolates of *B. henselae* SA2 strain (NCSU 95FO-099, Missy), *B. koehlerae* (NCSU 09FO-01, Trillium), and *B. quintana* (NCU11-MO-01 Monkey origin) were passed from agar plate-grown cultures into *Bartonella*-permissive cell lines, i.e., the DH82 (a canine monocytoid) cell line for strains *B. henselae* SA2, *B. quintana, B. vinsonii* subsp. *berkhoffii* I, and *B. koehlerae* and Vero cells (a mammalian fibroblast cell line) for *B. vinsonii* subsp. *berkhoffii* II and III to obtain antigens for IFA testing. For each antigen, heavily infected cell cultures were spotted onto 30-well Teflon-coated slides (Cell-Line/Thermo Scientific, Waltham, MA, USA), air-dried, acetone-fixed, and stored frozen. Fluorescein-conjugated goat anti-human immunoglobulin G (IgG) (Cappel, ICN, Costa Mesa, CA, USA) was used to detect bacteria within cells using a fluorescent microscope (Carl Zeiss Microscopy, LLC, Thornwood, NY, USA). Serum samples diluted in a phosphate-buffered saline (PBS) solution containing normal goat serum, Tween-20, and powdered nonfat dry milk to block nonspecific antigen-binding sites were tested with two-fold dilutions out to 1:8192. To avoid confusion with possible nonspecific binding found at low dilutions, a cutoff of 1:64 was selected as a seroreactive antibody titer.

### 4.4. Bartonella Enrichment Blood Culture/PCR Testing

*Bartonella* alpha-Proteobacteria growth medium (BAPGM) enrichment blood culture/PCR was performed as previously described. Briefly, qPCR and ddPCR amplifications were performed by targeting the 16S-23S intergenic transcribed spacer (ITS) region [30,34]. For qPCR products, amplicon identity was confirmed by DNA sequencing in a commercial laboratory (Genewiz, Inc, Research Triangle Park, NC, USA). Due to instrument design limitations, digital PCR droplets are not able to be sequenced [12]. Positive and negative control samples were consistently included when testing using both qPCR and ddPCR amplification modalities. Positive samples had bacterial loads equivalent to 1, 0.5, and 0.05 gene target copies per microliter. Negative controls consisted of DNA extracted from naïve human blood and the same molecular grade water used for all PCR preparations.

### 4.5. Statistics

Summary statistics were calculated for participant age and illness duration; distributions for each of these measures were examined visually for normality.

### 4.6. Ethics Approval and Consent to Participate

In accordance with the 1964 Helsinki Declaration, written informed consent to be a participant and to be part of a publication if printed was obtained from the study participant or parent (in the case of a minor). This study was carried out in accordance with all relevant guidelines applying to human study participation. No nonhuman animals were tested. The protocol was approved by the North Carolina State University (NCSU) Sponsored Programs and Regulatory Compliance, Institutional Review Board (IRB) (Approval IRB#1960). Specimen handling was performed following regional and national guidelines and regulations.

## Figures and Tables

**Figure 1 pathogens-09-01023-f001:**
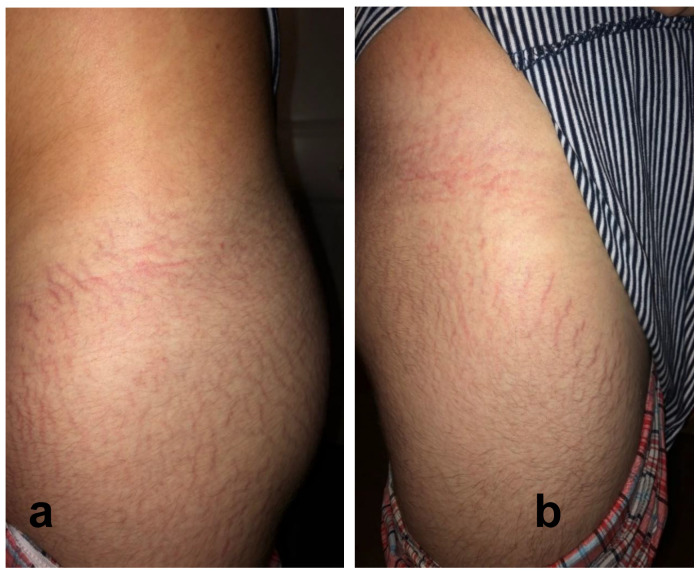
*Bartonella*-Associated Cutaneous Lesions. Fifteen-year-old male medically disabled student from North Carolina, with neuropsychiatric symptoms of six months’ duration. Serpiginous, vertical red lesions on the left hip and buttocks (**a**) and the right hip and upper thigh (**b**). The participant was positive for *Bartonella* infection by droplet digital PCR (ddPCR) enrichment blood culture (*Bartonella henselae* indirect fluorescent assay (IFA) serology titer 1:64). He reported no history of steroid use, bodybuilding, or rapid weight gain. Photograph of lesions provided by participant’s parents.

**Figure 2 pathogens-09-01023-f002:**
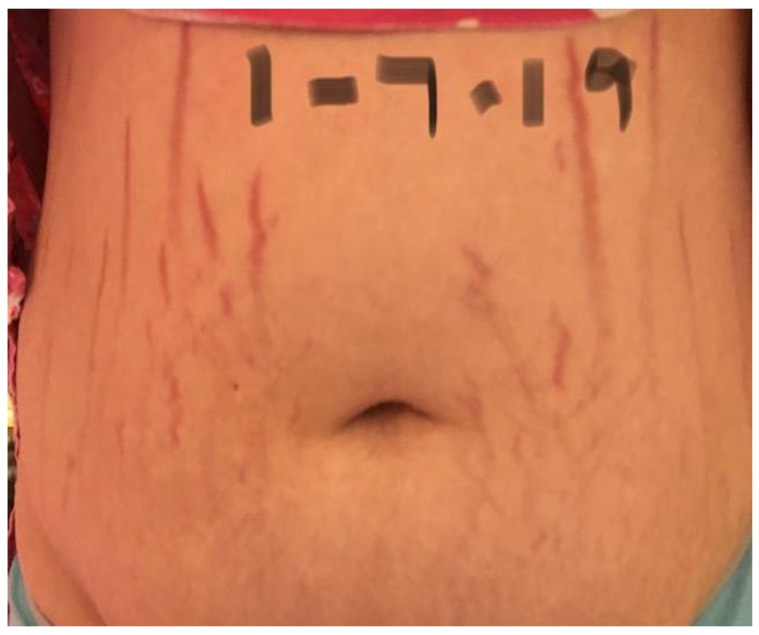
*Bartonella*-Associated Cutaneous Lesions. Sixteen-year-old female student from North Carolina, with neuropsychiatric symptoms of six years’ duration. Similar, but thinner and paler, but more serpiginous, vertical red lesions were photographed in November 2017. Positive for *Bartonella* infection by ddPCR enrichment blood culture (*Bartonella henselae* and *Bartonella koehlerae* IFA titers of 1:128 and 1:256, respectively). She reported no history of pregnancy or rapid weight gain. Photograph of abdominal lesions provided by participant’s parents.

**Figure 3 pathogens-09-01023-f003:**
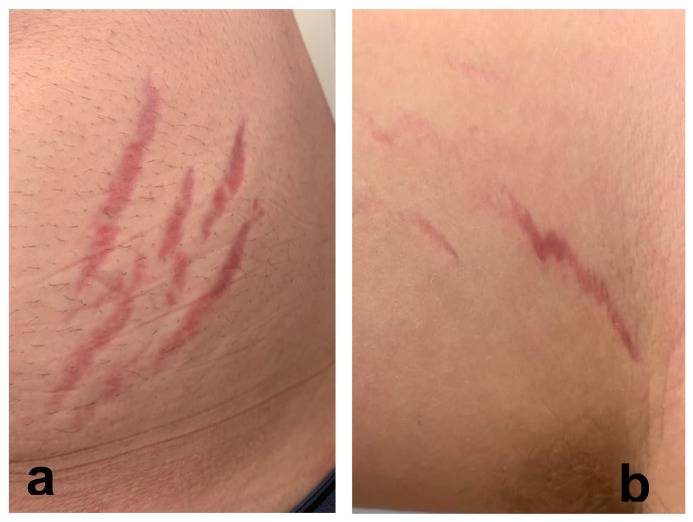
*Bartonella*-Associated Cutaneous Lesions. Seventeen-year-old male medically disabled student from North Carolina, with neuropsychiatric symptoms of one-year duration. Serpiginous, vertical red lesions on the left lower abdomen/upper groin (**a**) and the right axilla (**b**). Positive for *Bartonella* infection by ddPCR enrichment blood culture (Bartonella IFA serology-negative). He reported no history of steroid use, bodybuilding, or rapid weight gain. Photograph of lower abdomen, upper groin area and right axilla provided by participant’s parents.

**Figure 4 pathogens-09-01023-f004:**
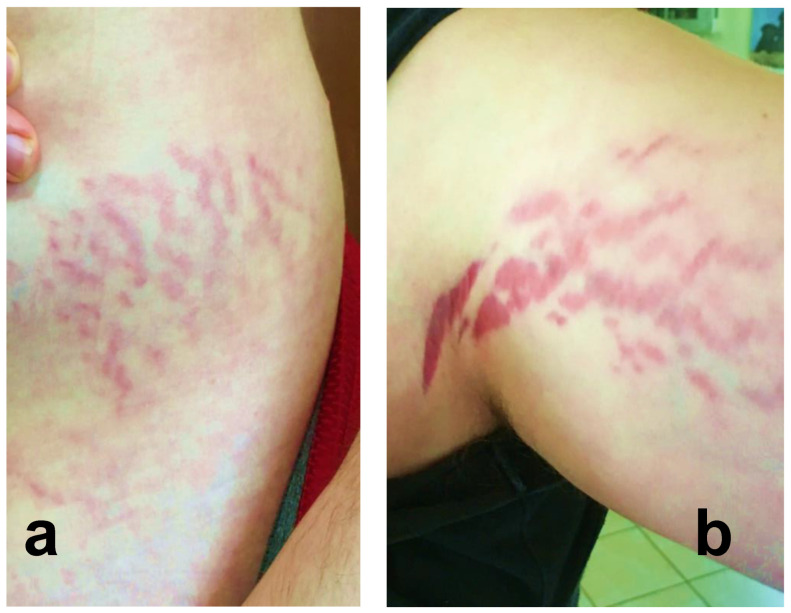
*Bartonella*-Associated Cutaneous Lesions. Twenty-year-old male medically disabled student from Virginia, with neuropsychiatric symptoms of four years’ duration. Serpiginous, vertical red lesions on the left hip (**a**) and on the left axilla (**b**). Positive for *Bartonella* infection by ddPCR enrichment blood culture (Bartonella IFA serology-negative). He reported no history of steroid use, bodybuilding, or rapid weight gain. Photographs of the hip and axilla provided by the study participant.

**Figure 5 pathogens-09-01023-f005:**
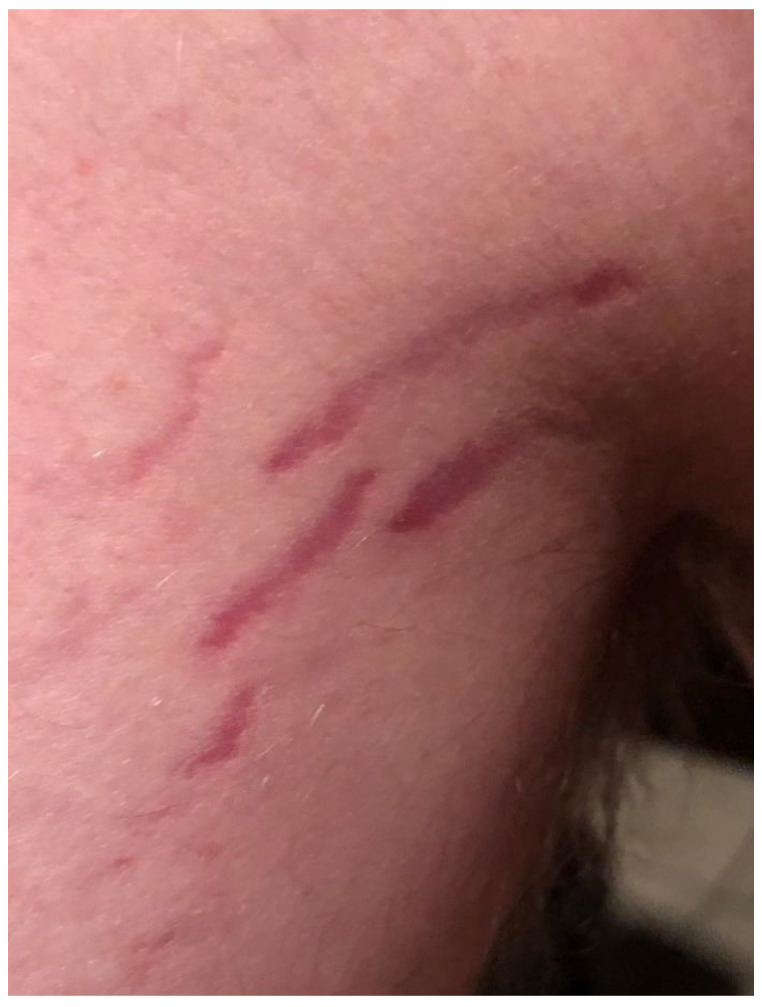
*Bartonella*-Associated Cutaneous Lesions. Twenty-one-year-old male landscaper from Connecticut, with neuropsychiatric symptoms of three years’ duration. Serpiginous, vertical red lesions on the right axilla. Positive for *Bartonella* infection by ddPCR enrichment blood culture (*Bartonella henselae* IFA titer 1:64). He reported no history of steroid use, bodybuilding, or rapid weight gain. Photograph of left axilla provided by the study participant.

**Figure 6 pathogens-09-01023-f006:**
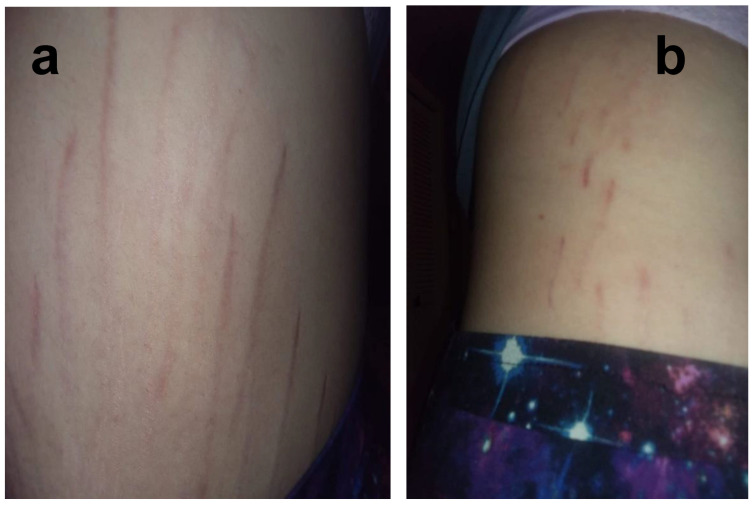
*Bartonella*-Associated Cutaneous Lesions. Twenty-one-year-old female veterinary assistant from Louisiana, with neuropsychiatric symptoms of 2 years’ duration. Serpiginous, vertical red lesions on the left abdomen (**a**) and the right upper side of the abdomen (**b**). Positive for *Bartonella* infection by ddPCR from direct EDTA blood sample and enrichment blood culture (*Bartonella henselae* and *Bartonella koehlerae* IFA titers of 1:256 and 1:64, respectively). Photo provided by the study participant.

**Figure 7 pathogens-09-01023-f007:**
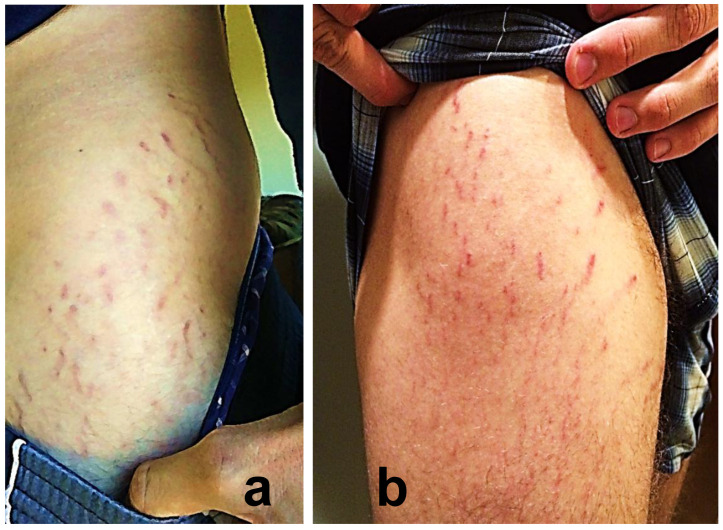
*Bartonella*-Associated Cutaneous Lesions. Twenty-two-year-old male medically disabled student from Minnesota, with neuropsychiatric symptoms of four years’ duration. Serpiginous, vertical red lesions on the left hip and buttocks (**a**) and the right upper thigh (**b**). Participant was positive for *Bartonella* infection by ddPCR enrichment blood culture (Bartonella IFA serology-negative). He reported no history of steroid use, bodybuilding, or rapid weight gain. Photographs of lesions provided by participant’s parents.

**Figure 8 pathogens-09-01023-f008:**
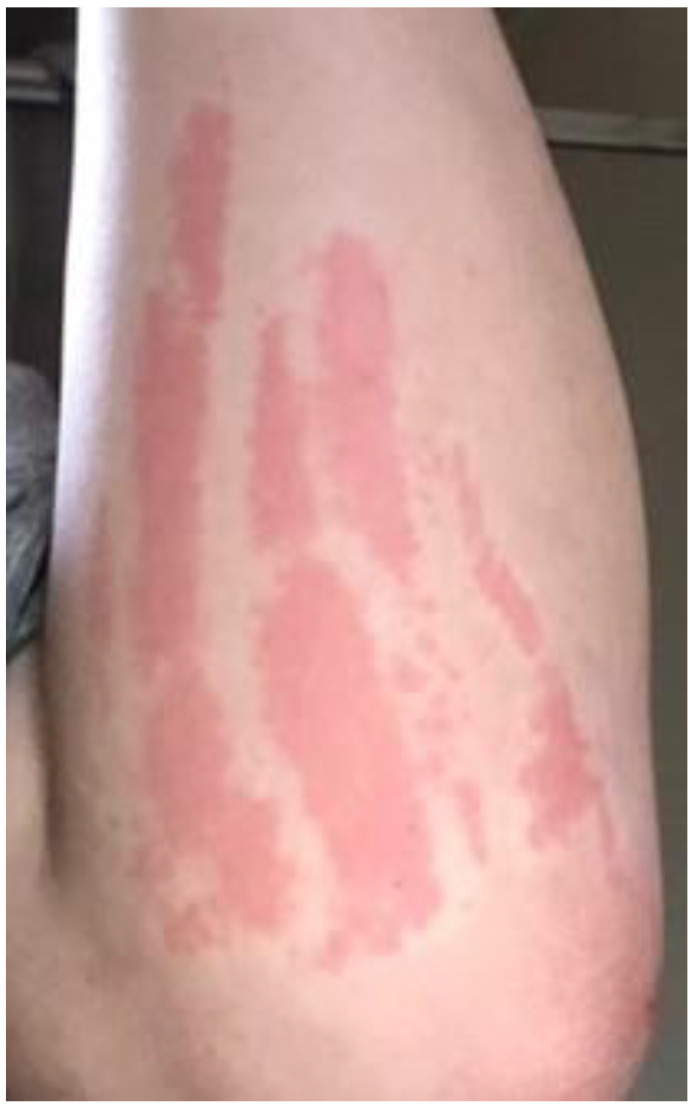
*Bartonella*-Associated Cutaneous Lesions. Twenty-one-year-old female medically disabled student from Mississippi, with neuropsychiatric symptoms of 13 years’ duration. Photograph of left forearm showing urticarial papules and plaques with dermatographism. Positive for *Bartonella* infection by ddPCR enrichment blood culture (Bartonella IFA serology-negative). Photo provided by the study participant.

**Figure 9 pathogens-09-01023-f009:**
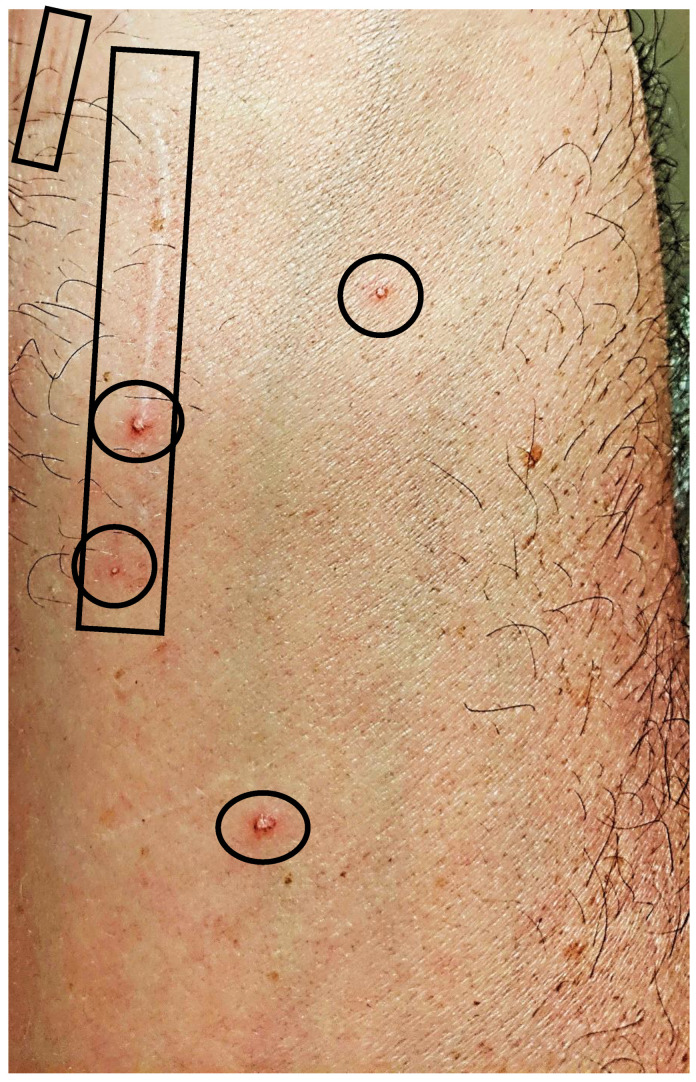
*Bartonella*-Associated Cutaneous Lesions. Thirty-four-year-old male veterinarian/rancher from Texas, with neuropsychiatric symptoms of six months’ duration. Photograph of forearm showing healed cat scratches (rectangles) (one year previously) and adjacent, recurrent pustular lesions (circles) that are not perifollicular. Positive for *Bartonella koehlerae* infection by qPCR and ddPCR (*Bartonella koehlerae* IFA titer 1:64). Photo provided by the study participant.

**Table 1 pathogens-09-01023-t001:** Symptoms checked off by participants on the Institutional Review Board-approved questionnaire, and self-reported conditions recorded by participants under the categories of prior diagnoses or treatments. Reported symptoms/conditions were divided into neuropsychological versus non-neuropsychological by the authors.

Neuropsychological Symptoms	Participants Reporting	Non-Neuropsychological Symptoms	Participants Reporting
Sleep Disorders ^	27	Fatigue ^	26
Mental Confusion ^	24	Ocular Involvement	20
Irritability/Rage ^	23	Eye Pain ^	15
Anxiety/Panic Attacks ^	22	Blurred Vision ^	15
Depression ^	21	Light Sensitivity *	4
Headache/Migraine ^	21	Floaters/Flashes *	2
Tremors ^	12	Joint/Back/Neck Pain ^	20
Hallucinations ^	7	GI Issues *	17
Institutionalized (Psych) *	2	Bowel Dysfunction ^	11
Postural Orthostatic Tachycardia (POTS) *	5	Diarrhea ^	8
Seizures ^	4	Abdominal Pain *	3
Schizophrenia *	2	Vomiting/Nausea ^	4
Suicidal *	2	IBS *	1
Tourettes *	2	Bloating *	1
Paralysis ^	2	Muscle Pain ^	15
Mast Cell Activation Syndrome *	2	Respiratory ^	15
MS-Like diagnosis *	1	Muscle Weakness ^	14
Sudden Onset OCD *	1	Numbness ^	13
PANS *	1	Tachycardia (Not POTS) ^	10
Psychosis *	1	Tendonitis ^	8
Stuttering *	1	Syncope ^	6
		Dental Issues ^	6
		Jaw Pain *	5
		Gingival Recession *	2
		Mouth Ulcers *	1
		Fever ^ of Unknown Origin *	5
		Ehrlos Danlos Syndrome *	3
		Cardiac (not reported elsewhere) *	2

^ Symptoms checked off by participants on the Institutional Review Board-approved questionnaire. * Participants provided written responses in the medical survey questionnaire indicating prior diagnoses and/or conditions for which treatments were administered. GI: gastrointestinal, IBS: irritable bowel syndrome, OCD: obsessive-compulsive disorder, PANS: Pediatric Acute-onset Neuropsychiatric Syndrome, and MS: multiple sclerosis.

**Table 2 pathogens-09-01023-t002:** *Bartonella* spp. serology and *Bartonella* alpha proteobacteria enrichment blood culture qPCR and ddPCR results for 29 individuals with self-reported neuropsychiatric illnesses.

	Reciprocal IFA Antibody Titers to:						Direct Blood PCR		BAPGM Enrichment Blood Culture/PCR	
Study No:	*Bartonella vinsonii berkhoffii* Type I	*Bartonella vinsonii berkhoffii* Type II	*Bartonella vinsonii berkhoffii* Type III	*Bartonella henselae* San Antonio 2	*Bartonella koehlerae*	*Bartonella quintana*	qPCR *	ddPCR	qPCR *	ddPCR
**1**	32	**64**	32	32	32	32	Neg	Neg	Neg	Neg
**2**	32	32	**64**	32	**64**	**128**	Neg	**Pos**	Neg	**Pos**
**3**	32	32	32	**64**	32	16	Neg	Neg	Neg	**Pos**
**4**	<16	32	32	**64**	16	<16	Neg	Neg	Neg	Neg
**5**	**256**	**64**	32	**128**	32	16	Neg	Neg	Neg	**Pos**
**6**	**128**	**128**	**64**	**64**	**128**	**128**	Neg	Neg	Neg	**Pos**
**7**	<16	**64**	**64**	**64**	<16	**64**	Neg	Neg	Neg	Neg
**8**	16	<16	<16	<16	<16	<16	Neg	Neg	Neg	**Pos**
**9**	<16	<16	16	<16	<16	16	Neg	**Pos**	Neg	**Pos**
**10**	<16	<16	<16	16	<16	<16	Neg	Neg	Neg	**Pos**
**11**	<16	**64**	**64**	**256**	**64**	32	Neg	**Pos**	Neg	**Pos**
**12**	<16	16	32	**64**	<16	32	Neg	Neg	Neg	**Pos**
**13**	<16	<16	<16	32	<16	<16	Neg	Neg	Neg	**Pos**
**14**	<16	<16	<16	<16	<16	<16	Neg	Neg	Neg	**Pos**
**15**	**64**	**64**	**64**	**128**	**64**	**64**	Neg	Neg	**Bh**	Neg
**16**	**256**	**128**	**256**	**128**	**64**	**128**	Neg	Neg	Neg	**Pos**
**17**	<16	32	32	**64**	<16	32	**Bh**	**Pos**	Neg	**Pos**
**18**	16	32	32	**64**	**64**	**64**	Neg	Neg	Neg	**Pos**
**19**	32	32	**64**	32	**64**	32	**Bk**	**Pos**	Neg	Neg
**20**	16	16	32	32	<16	**64**	Neg	Neg	Neg	Neg
**21**	32	16	32	**64**	32	32	Neg	Neg	Neg	**Pos**
**22**	<16	16	<16	16	<16	16	Neg	**Pos**	Neg	**Pos**
**23**	16	32	16	**64**	16	16	Neg	**Pos**	**Bh**	**Pos**
**24**	**256**	**64**	**64**	**64**	**64**	**64**	Neg	**Pos**	Neg	**Pos**
**25**	<16	<16	32	**64**	16	<16	Neg	Neg	Neg	Neg
**26**	<16	<16	32	**64**	32	16	Neg	Neg	Neg	Neg
**27**	<16	32	16	**128**	32	<16	Neg	Neg	Neg	Neg
**28**	**128**	**128**	**64**	**128**	**64**	**64**	Neg	Neg	Neg	**Pos**
**29**	<16	16	16	16	<16	16	Neg	Neg	Neg	**Pos**

* *Bartonella* species confirmed by DNA sequence analyses. IFA: indirect fluorescent assay, ddPCR: droplet digital PCR, and BAPGM: *Bartonella* alpha-Proteobacteria growth medium. Bold fonts were used to emphasize positive reults.

**Table 3 pathogens-09-01023-t003:** *Bartonella* ddPCR results generated using the *Bartonella* alpha-Proteobacteria growth medium (BAPGM) enrichment blood culture platform. All but one participant provided three blood samples obtained on alternate days for enrichment blood culture PCR testing. DNA was extracted from blood and from BAPGM enrichment blood cultures at 7, 14, and 21 days.

Participant	Source of DNA Tested by ddPCR			
	Blood	BAPGM Incubation Day 7	BAPGM Incubation Day 14	BAPGM Incubation Day 21
**1**	Neg	Neg	Neg	Neg
**2**	**Pos**	Neg	Neg	**Pos**
**3**	Neg	Neg	**Pos**	Neg
**4**	Neg	Neg	Neg	Neg
**5**	Neg	Neg	Neg	**Pos**
**6**	Neg	Neg	Neg	**Pos**
**7**	Neg	Neg	Neg	Neg
**8**	Neg	**Pos**	Neg	Neg
**9**	**Pos**	**Pos**	**Pos**	**Pos**
**10**	Neg	Neg	Neg	**Pos**
**11**	**Pos**	Neg	**Pos**	Neg
**12**	Neg	Neg	Neg	**Pos**
**13**	Neg	Neg	Neg	**Pos**
**14**	Neg	Neg	Neg	**Pos**
**15**	Neg	Neg	Neg	Neg
**16**	Neg	**Pos**	Neg	Neg
**17**	**Pos**	**Pos**	**Pos**	Neg
**18**	Neg	Neg	Neg	**Pos**
**19**	**Pos**	Neg	Neg	Neg
**20**	Neg	Neg	Neg	Neg
**21**	Neg	**Pos**	Neg	Neg
**22**	**Pos**	Neg	Neg	**Pos**
**23**	**Pos**	**Pos**	**Pos**	**Pos**
**24**	**Pos**	**Pos**	Neg	Neg
**25**	Neg	Neg	Neg	Neg
**26**	Neg	Neg	Neg	Neg
**27**	Neg	Neg	Neg	Neg
**28**	Neg	Neg	**Pos**	**Pos**
**29**	Neg	Neg	Neg	**Pos**

Bold fonts were used to emphasize positive reults.

## Data Availability

Data supporting the conclusions of the authors are included in the article. To assure participant confidentiality, please contact E.B.B. for questions relative to the raw data.

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
