# Peer review of "Bartonella Associated Cutaneous Lesions (BACL) in People with Neuropsychiatric Symptoms"

_pathogens, 2020, doi:10.3390/pathogens9121023_

Round 1

Author Response

Reviewer #1

Authors provided useful information of cutaneous lesions might be associated with Bartonella in the people with neuropsychiatric symptoms. They screened Bartonella infection in 33 patients with neuropsychiatric symptoms, 29 patients showed IFA and/or PCR positive. However, there are some issue to address.

We thank this reviewer for the overall positive assessment of our manuscript and for the very helpful suggestions that have improved upon the communication of our findings.

Major Comments Are there any reports of sero-prevalence of Bartonella infection in people without neuropsychiatric symptoms?

This is a good, but complex question. The answer is yes, but the qualifications are numerous. Seroprevalence using IFA serology has varied substantially among laboratories based upon the Bartonella species/strain used for testing, the population of individuals being tested and the number of IFA species/strains against which sera were tested. Reported seroprevalence in US blood donors was 3%, compared to a more recent study (both studies performed by CDC-Atlanta) where Bartonella henselae and Bartonella quintana  seroprevalence was 16 and 32% of 500 Brazilian blood donors tested by IFA, respectively. When we tested healthy volunteers from Duke University Medical School, only 1 of 32 individuals was B. henselae IFA seroreactive, when tested using the same IFA assay and the same research technician that performed testing in this study. In this study 23/29 (79%) participants were IFA seroreactive. Importantly, 22/29 individuals were qPCR or ddPCR positive, supporting infection at the time of sample collection, in contrast to serology which can only implicate exposure.  As suggested in the next comment, comparison of patient and controls are needed in future studies.

To more clarify association between Bartonella infection and symptoms, it has better to compare patient and healthy people.

Agreed. To date, obtaining funding to investigate the role of Bartonella spp. in association with various human disease processes has been challenging for investigators around the world. It is our hope that the publication of this data, in conjunction with publications from other research groups, will generate the reference data needed to support funded prospective studies.

According to the reference No. 34, the ddPCR has high sensitivity for detection of Bartonella. Could you describe more detail of the ddPCR and qPCR results? Both methods are able to provide quantitative information. Are there any relationship between amount of Bartonella and symptoms?

Samples that were qPCR/DNA sequence positive exhibited a Ct value ranging from 35.8 to 41.2, indicating a very low bacteria gene load in the four qPCR+ samples tested (these values represent between 1 and 0.05 bacteria gene target copy per microliter of sample). Samples positive by ddPCR also showed a very low bacteria gene load.  The number of positive drops for Bartonella varied between 1 and 3 per sample. This information has been added to the results section.  We have also added a couple of sentences to the first paragraph of the discussion to better orient the reader to ddPCR as a more sensitive diagnostic testing modality.

Due to the very low bacterial load and the information obtained via the questionnaire, it is not possible to address a relationship between the amount of Bartonella and patient symptoms. This will be interesting to assess in future studies looking at other patient populations.

Author did sequence of amplicon from the qPCR, it is better to provide the accession number of that sequences. If sequences were analysed by BLAST search, please provide the results also.

We have provided the sequence identities for the BLAST search for the B. henselae and B. koehlerae qPCR amplicons sequenced in this study. 

Did you put positive control and negative control for these PCRs?

Yes, positive and negative controls were routinely run with each qPCR and ddPCR reaction. We have added to the methods: Positive and negative control samples were always included when testing using both PCR amplification modalities.  Positive samples had bacterial loads equivalent to 1, 0.5 and 0.05 gene target copies per microliter. Negative controls consisted of DNA extracted from naïve human blood and the same molecular grade water used for all PCR preparations.

We have also stated in the results” All qPCR and ddPCR negative controls remained negative throughout the study.

Title “Bartonella Associated Cutaneous Lesions (BACL) cutaneous lesions in people with neuropsychiatric symptoms” In the title, words “cutaneous lesions” are overlapped. Thus, just “Bartonella Associated Cutaneous Lesions (BACL) in people with neuropsychiatric symptoms” is fine.

Modified as suggested.

It is a little bit complicated to understand the results only read the sentence. Please summarize the results in a table for easy understanding.

We very much appreciate this suggestion. We have added two tables that provide additional detail relative to the reported results.

The position of the period is wrong though out the manuscript. Actually these issues are so many in the manuscript. For example, Line 42: “…onset illness caused by Bartonella henselae.[1,2] However…” This should be “…onset illness caused by Bartonella henselae [1,2].

Corrected throughout

However…” Line 110 Change “…and plaques with dermatographism. (Figure 7).” to “…and plaques with dermatographism (Figure 7).”

Modified as suggested.

Minor comments Line 26 “…Review Board (IRB). Approval IRB#1960.” Change to “…Review Board (IRB) (Approval IRB#1960).”

Modified as suggested.

Line 37 “bartonella” should be “Bartonella”.

Corrected

Line 78: “manuscript” should be changed to “study” or “investigation”.

Changed to investigation.

Line 115 “…(Figure 1A) and the right hip and upper thigh (Figure 1B).” you use small character with in the figure. Thus, change to “…(Figure 1a) and the right hip and upper thigh (Figure 1b).”

Same issue is observed in Figure 3, 4, 6, and 8.

Line 304-305 “BAPGM (Bartonella alpha proteobacteria growth medium) enrichment blood culture/PCR 304 was performed as previously described.” Please put reference article for this sentence.

Line 256 “bloodstream PCR” should be need space “blood stream PCR” There are scattered places that seem to be double spaces. Thus, please check carefully.

Modified and checked throughout.

Reviewer 2 Report

After reviewing the manuscript entitled “Bartonella Associated Cutaneous Lesions (BACL) cutaneous lesions in people with neuropsychiatric symptoms”, Breitschwerdt et al. I only have only minor comments.

The authors should explain in more detail why they included a group of patients with neuropsychiatric symptoms in their research? Do the authors expect that Bartonella infection may predispose to the development or worsening of these symptoms?

The authors adopted an unprecedented way of citing in the text, e.g. “….caused by Bartonella henselae. [1,2] However, a 43 cutaneous primary inoculation papule is also found in 60 to 90% of CSD cases. [3] Although rare….” I checked other papers published in Pathogens and found no such citation style. I suggest the authors adapt to the journal's requirements.

L26 “…..Institutional Review Board (IRB).  Approval ….” - double space, similar L24,  L37, L72, L85, L91

Author Response

Reviewer #2

After reviewing the manuscript entitled “Bartonella Associated Cutaneous Lesions (BACL) cutaneous lesions in people with neuropsychiatric symptoms”, Breitschwerdt et al. I only have only minor comments.

We appreciate this reviewers suggestions relative to our manuscript.

The authors should explain in more detail why they included a group of patients with neuropsychiatric symptoms in their research? Do the authors expect that Bartonella infection may predispose to the development or worsening of these symptoms?

We appreciate this question. Both possibilities proposed by the reviewer are feasible. We have purposefully attempted to not overstate our findings and in the discussion we clearly state that detection of Bartonella spp. DNA in these patients does not confirm causation for the BACL or the neuropsychiatric symptoms. We have also emphasized the need for future studies to differentiate these two possibilities. Based upon our PANS case report, we do believe that Bartonella can be causative in a subset of infected individuals.

The authors adopted an unprecedented way of citing in the text, e.g. “….caused by Bartonella henselae. [1,2] However, a 43 cutaneous primary inoculation papule is also found in 60 to 90% of CSD cases. [3] Although rare….” I checked other papers published in Pathogens and found no such citation style. I suggest the authors adapt to the journal's requirements.

We have formatted the manuscript to adhere to the journal’s requirements.

L26 “…..Institutional Review Board (IRB).  Approval ….” - double space, similar L24,  L37, L72, L85, L91

Corrected in the revised manuscript.

Round 2

Reviewer 1 Report

Authors provided useful information of cutaneous lesions might be associated with Bartonella in the people with neuropsychiatric symptoms. They screened Bartonella infection in 33 patients with neuropsychiatric symptoms, 29 patients showed IFA and/or PCR positive. The manuscript is well written. However, some minor issues are should be addressed.

Minor comments

Line 48-49: Line breaks in the middle of the sentence

Line 50 “bloodstream” change to “blood stream”

Line 242-249 These sentences were completely match with the sentences within the reference (no. 12). It seemed to be taken from that reference. Authors have to do modify these sentences or put quotation marks to indicate direct quote.

Line 336-337 The font of the characters is different.

Author Response

Answers to Reviewer 1:

Minor comments

Point 1 L48-49: Line breaks in the middle of the sentence

Response 1: Line break mid-sentence has been removed.

Point 2 L50:  “bloodstream” change to “blood stream”

Response 2: Bloodstream has been changed to blood stream.

Point 3 L242-249: These sentences were completely match with the sentences within the reference (no. 12). It seemed to be taken from that reference. Authors have to do modify these sentences or put quotation marks to indicate direct quote.

Response 3: The reason that these sentences are a match to reference 12 is because the referenced paper was written by our laboratory and this is our standard description of the ddPCR methodology.

Point 4 L336-337: Line 336-337 The font of the characters is different.

Response 4: Font has been changed to Palatino Linotype to match the reformatted text style decided upon by the journal.